# Novel Survival Features Generated by Clinical Text Information and Radiomics Features May Improve the Prediction of Ischemic Stroke Outcome

**DOI:** 10.3390/diagnostics12071664

**Published:** 2022-07-08

**Authors:** Yingwei Guo, Yingjian Yang, Fengqiu Cao, Wei Li, Mingming Wang, Yu Luo, Jia Guo, Asim Zaman, Xueqiang Zeng, Xiaoqiang Miu, Longyu Li, Weiyan Qiu, Yan Kang

**Affiliations:** 1College of Medicine and Biological Information Engineering, Northeastern University, Shenyang 110169, China; 1910442@stu.neu.edu.cn (Y.G.); 1810453@stu.neu.edu.cn (Y.Y.); 1510549@stu.neu.edu.cn (F.C.); 20183290060@stumail.sztu.edu.cn (X.M.); 2College of Health Science and Environmental Engineering, Shenzhen Technology University, Shenzhen 518118, China; liwei2@sztu.edu.cn (W.L.); zamanasim2021@email.szu.edu.cn (A.Z.); 2100411016@stumail.sztu.edu.cn (X.Z.); 201905100203@stumail.sztu.edu.cn (L.L.); 201905100206@stumail.sztu.edu.cn (W.Q.); 3Department of Radiology, Shanghai Fourth People’s Hospital Affiliated to Tongji University School of Medicine, Shanghai 200434, China; wangmingming000@126.com; 4Department of Psychiatry, Columbia University, New York, NY 10027, USA; jg3400@columbia.edu; 5Engineering Research Centre of Medical Imaging and Intelligent Analysis, Ministry of Education, Shenyang 110169, China; 6School of Applied Technology, Shenzhen University, Shenzhen 518060, China

**Keywords:** ischemic stroke outcome, clinical text information, radiomics features, survival features, machine learning

## Abstract

Background: Accurate outcome prediction is of great clinical significance in customizing personalized treatment plans, reducing the situation of poor recovery, and objectively and accurately evaluating the treatment effect. This study intended to evaluate the performance of clinical text information (CTI), radiomics features, and survival features (SurvF) for predicting functional outcomes of patients with ischemic stroke. Methods: SurvF was constructed based on CTI and mRS radiomics features (mRSRF) to improve the prediction of the functional outcome in 3 months (90-day mRS). Ten machine learning models predicted functional outcomes in three situations (2-category, 4-category, and 7-category) using seven feature groups constructed by CTI, mRSRF, and SurvF. Results: For 2-category, ALL (CTI + mRSRF+ SurvF) performed best, with an mAUC of 0.884, mAcc of 0.864, mPre of 0.877, mF1 of 0.86, and mRecall of 0.864. For 4-category, ALL also achieved the best mAuc of 0.787, while CTI + SurvF achieved the best score with mAcc = 0.611, mPre = 0.622, mF1 = 0.595, and mRe-call = 0.611. For 7-category, CTI + SurvF performed best, with an mAuc of 0.788, mPre of 0.519, mAcc of 0.529, mF1 of 0.495, and mRecall of 0.47. Conclusions: The above results indicate that mRSRF + CTI can accurately predict functional outcomes in ischemic stroke patients with proper machine learning models. Moreover, combining SurvF will improve the prediction effect compared with the original features. However, limited by the small sample size, further validation on larger and more varied datasets is necessary.

## 1. Introduction

Ischemic stroke is the primary reason for disability and the second leading cause of death worldwide [1]. Acute ischemia is caused by a sudden occlusion of the arteries in the brain [2], leading to a penumbra or core infarct in the brain tissue [3]. Thus, surviving patients are usually accompanied by varying neurological deficits, resulting in impaired living quality and burdened families and society.

The primary treatments to salvage damaged tissue are thrombolysis and mechanical thrombectomy [4]. Appropriate therapeutic strategies will bring the optimal treatment effect and functional recovery; however, there may be certain risks in the treatment process [5]. Therefore, the risks of suffering from poor prognosis are as crucial as the expected therapeutic effect in designing treatment plans. If the therapeutic effects and outcomes under different treatments can be predicted before the treatment, the outcome risks will be effectively reduced. Accurate outcome prediction will assist physicians in customizing personalized treatment plans, reducing the situation of poor recovery, and objectively and accurately evaluating the treatment effect [6]. However, patients always display heterogeneity and uncertainty of the treatment benefit. Therefore, selecting patients who will benefit from thrombolysis treatment during the decision-making process is challenging.

Clinically, physicians usually evaluate stroke outcomes based on their experience, which requires superb professional skills and rich clinical experience. However, due to limitations affecting the varying professional levels, it is difficult to ensure a high prediction accuracy, which influences the selection of treatment methods and the management of rehabilitation plans. Therefore, many scholars and companies are gradually focusing on researching and developing objective and accurate outcome prediction tools [7,8]. However, it is difficult to meet clinical needs due to prediction accuracy. The main reason is that clinical text information (CTI) and imaging information have not been effectively combined to improve prediction accuracy [9].

Several studies have shown that stroke outcomes correlate with CTI and stroke parameters computed from images. Lea-Pereira et al. [10] predicted mortality risk scores during admission for ischemic stroke with CTI such as age, sex, readmission, and neurological symptoms, and an AUC of 0.742 was obtained with a logistic regression model. Ref. [11] concluded the correlation between basic patient information in CTI and long-term stroke recurrence. Xie et al. [12] used patient information (age and gender), clinical scores (ASPECT, NIHSS, HMACS, NASCET, and TIMI) obtained from multimodal images (CT, CTP, and CTA), and volumes of lesion tissue obtained from CTP to predict the modified Rankin score (mRS) in three months. A maximum AUC of 0.873 was achieved. Moreover, Brugnara et al. [13] combined additional location information for lesions, hypertension, diabetes, dizziness, and physical symptoms and achieved a similar AUC of 0.856. Alaka et al. [14] adopted machine learning models to select prominent features from CTI for outcome prediction, and the highest AUC was 0.71. Furthermore, some scholars have carried out relevant studies with medical images to explore the relationship between outcomes and neuroimaging. Osama et al. [15] applied the parallel multiparametric feature embedded Siamese network (PMFE-SN) to perform multiclassification prediction using CBF, CBV, TTP, Tmax, and MTT in the ISLES 2017 datasets [16]. This method obtained an ACC of 0.64. Choi et al. [17] predicted mRS using three different models (convolutional neural network, logistic regression, and the integration of both) with the ISLES 2016 datasets and obtained a mean absolute error of 1.37 ± 1.00, 1.26 ± 0.81, and 1.10 ± 0.70, respectively. In addition, some scholars have performed outcome prediction based on other types of information. For example, Ref. [18] used the neutrophil–lymphocyte ratio to predict mRS and achieved an AUC of 0.88. Although clinical variables have consistently been associated with outcomes after ischemic stroke, the usefulness of neuroimaging in predicting outcomes has not been definitively established [9].

Imaging information is crucial for predicting ischemic stroke outcomes. For imaging feature extraction, radiomics is an innovative method to quantify high-dimensional features from medical images. At present, radiomics is widely used in cancer and tumors and has achieved excellent results. For example, it is used to investigate tumor heterogeneity [19,20] and in clinical decision support systems to improve treatment decision making and accelerate advancements of clinical decision support systems in cancer medicine [21,22,23,24,25,26]. However, in the field of stroke, only a few studies have explored the role of radiomics in diagnosing ischemic stroke [27], penumbra-based prognosis assessment [28], and functional prediction [29]. For example, Tang et al. [28] used a Lasso model to achieve multiclassification prediction with a maximum AUC of 0.77 by using radiomics features extracted from infarct and penumbra in CBF and ADC. Although previous researchers have verified the critical value of DSC-PWI perfusion sequences in stroke diseases, few studies have explored the relationship between DSC-PWI imaging characteristics in the temporal dimension and prognosis. However, the correlation between this information and functional recovery is worth further exploration due to sufficient blood flow information. Furthermore, other information, such as electroencephalography [30] and brainstem auditory evoked potential [31], has been used to predict neurological recovery in stroke patients.

Based on previous research, there are still some deficiencies in predicting stroke outcomes:(A)Accurate multilevel prediction methods are expected. Most proposed risk prediction models only perform prediction of two situations, good outcome (mRS ≤ 2) and poor outcome (mRS > 2). Compared with the prediction of two situations, multilevel prediction of outcomes scores (mRS ranging from 0 to 6) will undoubtedly provide more targeted assistance to the clinic, such as formulating appropriate therapeutic strategies and predicting the achieved recovery.(B)The value of CTI and image features in prognostic prediction needs to be further explored. Previous studies used CTI from clinical records or image information from medical images to predict patient recovery. However, it is difficult to fully express patients’ overall state and disease development tendency using single-dimensional information. Therefore, exploring the combined role of CTI and imaging information in stroke prognosis is necessary. In addition, it is also of great significance in developing powerful features associated with functional recovery generated by the fusion of the two.(C)An analysis method based on cerebral blood flow dynamic characteristics is also lacking. Previous studies used CBF, CBV, Tmax, MTT, and TTP images computed from perfusion sequence for outcome prediction. However, these images could only obtain static parameters, making it difficult to reflect the dynamic information in blood flow transmission. Moreover, missing information may be precisely hidden in the perfusion sequence. Therefore, an analysis method should be proposed based on cerebral blood flow dynamic characteristics.

This study provides a prediction method of stroke outcome to make up for the above deficiencies. Detailed outcome scores were predicted by combining CTI and dynamic blood flow characteristics in DSC-PWI. It may provide a scientific research foundation and clinical assistance for precision medicine.

## 2. Materials and Methods

Detailed materials and methods are introduced in the following subsections. The methods in this study include: (1) preprocessing DSC-PWI datasets and making regions of interest (ROIs), (2) calculating radiomics features, (3) selecting outstanding features and selecting the excellent feature selection method, and (4) multidimensional feature fusion and predicting ischemic stroke outcomes.

### 2.1. Materials

This retrospective study was approved by the Institutional Review Boards of Shanghai Fourth People’s Hospital Affiliated to Tongji University School of Medicine and exempted from informed consent. The datasets in our study were collected by the neurology department of Shanghai Fourth People’s Hospital Affiliated to Tongji University School of Medicine, China, from 2013 to 2016. A total of 80 DSC-PWI images from 56 patients with ischemic stroke were retrospectively reviewed and included. All patients were imaged within 24 h of symptom onset, and 22 patients were screened at least twice during pretreatment and post-treatment. The primary clinical information recorded as CTI includes the age, sex, income NHISS, outcome NHISS, right limbs weakness, left limbs weakness, lisp out, confusion, hypertension, diabetes, atrial fibrillation, and volumes of stroke lesions. The DSC-PWI image for each patient was scanned on a 1.5T MR scanner (Siemens, Germany), and Table 1 shows the details.

### 2.2. Methods

Figure 1 shows a flowchart of this study’s methods. Specifically, the proposed method in this study includes four steps: preprocessing datasets and making ROIs, calculating radiomics features, selecting outstanding features and excellent feature selection method, multidimensional feature fusion, and predicting ischemic stroke outcomes. The following process is a detailed description of the methods.

#### 2.2.1. Preprocessing DSC-PWI Datasets and Making ROIs

Preprocessing of datasets is intended to reduce noise and position deviation impacts. First, this study corrected DSC-PWI datasets for potential patient motion by registering all the volumes in the time series with the multiplicative intrinsic component optimization algorithm [31,32]. Then, data smoothing filtering was performed to decrease the noise interference generated by the equipment and other factors in the DSC-PWI images. In detail, the triple moving average filter was selected to smooth the data voxel-by-voxel with a 1 × 3 filtering kernel. As a result, in the DSC-PWI datasets, the intensity of each pixel in the time dimension formed a time–intensity sequence I(t), which was used in the subsequent analysis.

In addition, the necessary condition for comparative analysis of normal and abnormal cerebral blood flow is to detect both locations accurately. This study used a fully automated Rapid Processing of Perfusion and Diffusion (RAPID) software (iSchemaView, Menlo Park, CA, USA) [33] to segment the lesion tissue (LT) of ischemic stroke with the condition of Tmax >6 s [34]. In contrast with the LT, the non-lesion tissue in the symmetrical region of LT was defined as normal tissue (NT). Finally, 80 sets of ROIs for LT and NT were generated from the DSC-PWI datasets.

#### 2.2.2. Calculating Radiomics Features

Some studies [35,36,37,38] have shown that the time–intensity curve of LT in DSC-PWI images with ischemic stroke has a much smaller brightness decrease than the curve of NT. Therefore, the data of DSC-PWI in the time dimension are correlated with the blood flow state of the brain tissue to a certain extent. This study used radiomics technology to compute high-dimensional image features based on the LT ROIs and NT ROIs in the DSC-PWI images. In detail, the DSC-PWI datasets are four-dimensional (4D) images composed of N three-dimensional (3D) images with a size of S × H × W, where N is the total number of 3D images in the temporal dimension, and S, H, and W represent the slice, height, and width of the 3D images, respectively. First, by decomposing the 4D images into N (50 in this study) single 3D images, the radiomics features for 3D images could be computed separately at each time. Then, 65,800 radiomics features (50 3D images × 1316 features) could be calculated from each DSC-PWI image. These radiomics features were divided into nine groups: (1) Shape-based (Shape, 14 features × 50 = 700 features), (2) First-Order Statistics (First-order, 18 features × 50 = 900 features), (3) Gray-Level Co-Occurrence Matrix (GLCM, 24 features × 50 = 1200 features), (4) Gray-Level Run-Length Matrix (GLRLM, 16 features × 50 = 800 features), (5) Gray-Level Size-Zone Matrix (GLSZM, 16 features × 50 = 800 features), (6) Neighboring Gray-Tone Difference Matrix (NGTDM, 5 features × 50 = 250 features), (7) Gray-Level Dependence Matrix (GLDM, 14 features × 50 = 700 features), (8) Log-Sigma (Log-Sigma, 465 features × 50 = 23,250 features), and (9) Wavelet-based (Wavelet, 744 features × 50 = 37,200 features). Radiomics feature calculation was automatically performed using the PyRadiomics package implemented in Python [39,40]. In this study, each 3D image in the DSC-PWI data was defined as S(n), n∈[0, 49], and the DSC-PWI image was represented as set {S(0), S(1), …, S(49)}. Moreover, the calculated features were renamed by connecting their original name and the n-value of the 3D image S(n). For example, “log-sigma-1-0-mm-3D_firstorder_Skewness_17” represents the radiomics feature “log-sigma-1-0-mm-3D_firstorder_Skewness” of S(17), which is the 17th 3D image in DSC-PWI data, and this feature belongs to the Log-Sigma group.

#### 2.2.3. Selecting Outstanding Lesions Features and Excellent Feature Selection Method

First, this study applied multilevel selection strategies to obtain outstanding temporal features representing stroke lesions. Then, the optimal feature selection method was determined based on the score of selected features on multiple learning models. Finally, the selected method and lesion features were combined with CTI to predict ischemic stroke outcomes.


(A)Selecting Significant Features


A *t*-test analysis was performed to screen the significant features between LT and NT. First, a normalization operation was centered on the mean and scaled to unit variance according to Equation (1).
(1)$$F_{i}^{*}=(F_{i}-\bar{F_{i}})/\left(F_{imax}-F_{imin}\right)$$
where $F_{i}^{*}$ is the normalized feature of the i*th* feature $F_{i}$, and the variables $\bar{F_{i}}$, $F_{imax}$, and $F_{imin}$ are the mean, maximum, and minimum of *F_i_*, respectively.

Before the *t*-test analysis, a homogeneity of variance test was performed to detect homogeneity of variance between the LT and NT features. When the feature had homogeneity of variance, the *t*-test was performed directly. However, if the feature did not have homogeneity of variance, the parameter *equal_val* = *False* needed to be added during the *t*-test analysis (seen in Figure 2). The homogeneity of variance test method used was the Levene test. Finally, the significant features with values of *p* < 0.05 in the *t*-test analysis remained to complete subsequent feature selection processing.


(B)Selecting Feature Sets Based On Multiple Feature Selection Methods


Feature selection aims to find the most compelling feature representing the target variable and compress the feature space. This study used 13 feature selection methods based on diversity principles to select features from significant features. These 13 methods perform feature extraction from mutual information, information entropy, cluster statistics, sparse feature, linear relation, and other aspects, respectively (seen in Table 2). In detail, the feature selection methods contain four types: the first type is based on theoretical information (TI), including conditional mutual information maximization (CMIM), joint mutual information (JMI), mutual information feature selection (MIFS), mutual information maximization (MIM), and minimal redundancy maximum relevance (MRMR); the second type is based on similarity features (SIF), including Fisher score (Fisher), Lap score (LS), and ReliefF); the third type is based on statistical features (STF), including F-score (FS) and T-score (TS); and the final type is based on steaming and sparse learning (SSL), including multicluster feature selection (MCFS), Alpha investing (Alpha), and the least absolute shrinkage and selection operator (Lasso). The above methods were introduced in Refs. [41,42,43,44,45,46] and implemented in Python 3.6.

When performing feature selection, the target variables were defined as 0 and 1 to represent NT and LT. Then, the 13 methods were applied to select the matched features from the significant features. Moreover, during the implementation, while Lasso selected features with coefficients more prominent than 0.02 to control the number of features within the set maximum feature length of 20, the others obtained features whose score exceeded 0.9, and the total number was less than 20. Moreover, the features extracted from them were regarded as feature set $F_{method}$, where *method* represents the name of the feature selection method (CMIM, JMI, MIFS, MIM, MRMR, Fisher, LS, ReliefF, MCFS, Alpha, Lasso, FS, TS).

**Table 2 diagnostics-12-01664-t002:** Descriptions of the 13 feature selection methods in this study.

Type	Method	Description	Equation
FITI	MIM	Evaluates features by correlation between features and classes measured by mutual information	$MIM(f_{i})=I(f_{i};C)$
	MIFS/MRMR	Evaluates features by correlation between features and classes and redundancy among features	$MIFS(f_{i})=I(f_{i};C)-\beta \sum_{s_{j}\in S}I(f_{i};f_{S})$ $MRMR(f_{i})=I(f_{i};C)-\frac{1}{S}\sum_{s_{j}\in S}I(f_{i};f_{S})$
	JMI/CMIM	Evaluates features by correlation between features and classes and redundancy among features measured by conditional mutual information	$JMI(f_{i})=I(f_{i};C)-\frac{1}{\left|S\right|}\sum_{s_{j}\in S}\left[I(f_{i};C)-I(f_{i};C|f_{S}\right)]$ $CMIM(f_{i})=\underset{f_{s}\in S}{\min}I(f_{i};C|f_{s})$
SIF	Fisher/LS	Compares features with their ratios of variance between classes and variance within classes	$\mathrm{Fisher}(k)=\frac{R_{B}^{\left(k\right)}}{R_{w}^{\left(k\right)}}$ $LS(f_{i})=\frac{\sum_{ab}{\left(f_{ra}-f_{rb}\right)}^{2}W_{ij}}{Var(f_{r})}$
	ReliefF	Compares features with correlation between features and classes computed from ability of features to distinguish between close samples	$ReliefF(f_{i},R1,R2)=\frac{\left|R1(A)-R2(A)\right|}{\max(A)-\min(A)}$
STF	FS	Obtains feature score with ability to distinguish positive classes and negative classes computed by average of both classes	$FS(i)=\frac{{\left({{\bar{f}}_{i}}^{\left(+\right)}-{\bar{f}}_{i}\right)}^{2}+{\left({{\bar{f}}_{i}}^{\left(-\right)}-{\bar{f}}_{i}\right)}^{2}}{\frac{1}{n_{+}-1}\sum_{k=1}^{n_{+}}{\left({f_{k,i}}^{\left(+\right)}-{{\bar{f}}_{i}}^{\left(+\right)}\right)}^{2}+\frac{1}{n_{-}-1}\sum_{k=1}^{n_{-}}{\left({f_{k,i}}^{\left(-\right)}-{{\bar{f}}_{i}}^{\left(-\right)}\right)}^{2}}$
	TS	Computes feature score with average and variance of features	$TS(i)=\frac{\left({{\bar{f}}_{i}}^{\left(+\right)}-{{\bar{f}}_{i}}^{\left(-\right)}\right)}{\frac{1}{n_{+}-1}\sum_{k=1}^{n_{+}}{\left({f_{k,i}}^{\left(+\right)}-{{\bar{f}}_{i}}^{\left(+\right)}\right)}^{2}+\frac{1}{n_{-}-1}\sum_{k=1}^{n_{-}}{\left({f_{k,i}}^{\left(-\right)}-{{\bar{f}}_{i}}^{\left(-\right)}\right)}^{2}}$
SSL	MCFS	Combines cluster with feature coefficients of combinatorial classes to compute feature score	$MCFS(i)=\frac{\max}{k}|f_{k,i}|$
	Alpha	Evaluates features by dynamically adjusting threshold on error reduction to obtain selection results	$E(N_{i})/E(M_{i})<\alpha_{\Delta }/(1-\alpha_{\Delta })$
	Lasso	Uses L1 regularization to make weight of some learned features equal 0, to achieve purpose of sparse and feature selection	$Lasso(\overset{\wedge }{\beta })=arg\;min\left\{\sum_{i=1}^{n}{\left(y_{i}-\beta_{0}-\sum_{j=1}^{p}\beta_{j}x_{ij}^{*}\right)}^{2}+\lambda \sum_{j=0}^{p}\left|\beta_{j}\right|\right\}$


(C)Selecting Outstanding Lesion Features and the Best Feature Selection Method


Given that the same features may perform differently on various models, this study applied ten supervised machine learning models to rank the 13 *F_method_* to find the best feature selection method. In detail, the composite score (CS) defined as Equation (2) was applied to evaluate the classification ability and robustness. The CS was obtained from five indexes, including the area under the curve score (Auc), accuracy (Acc), precision (Pre), F1-score (F1), and Recall of the machine learning model. The CS of the feature set was the average of the five indexes on the ten models. The ten machine learning models include support vector machine (SVM), decision tree (DT), Adaboost classifier (Ada), neural network (NN), random forest (RF), k-nearest neighbors (KNN), logistic regression (LR), linear discriminant analysis (DA), gradient boosting classifier (GBDT), and GaussianNB (NB) (seen in Table 3).

This study used the 13 *F_method_* to perform tenfold cross-validation on the ten learning models for computing the Pre, Acc, Auc, F1, and Recall. Then, the average of the five indexes on the ten learning models was defined as the CS. The CS value not only reflects the average result of five tenfold cross-validation of each feature set on the same model but also reflects the overall ability of the feature set on the ten models. Therefore, this study defined the features extracted from the 13 methods as the outstanding lesion features. Moreover, the method that extracts the feature set with the best CS was regarded as the most prominent feature selection method (*best_method*).
(2)$$CS(F_{method})=\frac{1}{KM}\sum_{k,m}index(k,model(m,F_{method}))$$
where *K* and *M* are the total indexes and learning models, respectively. *K* = 5, *M* = 10, *k*∊{Pre, Acc, Auc, F1, Recall}, m∊{SVM, DT, Ada, NN, RF, KNN, LR, DA, GBDT, NB}. *index* (*k*, *model*(*m*, *F_method_*)) represents the k*th* index of the m*th* model fitted by *F_method_*.

Depending on the 13 *F_method_* and ten models, 130 (13 × 10) classifiers were obtained. These classifiers were defined by combining the learning machine model and the feature selection method. For example, *C_SVM_MIM_* represents the classifier fitted by SVM and feature sets *F_MIM_*.

#### 2.2.4. Multidimensional Feature Fusion

After obtaining the 13 *F_method_* that could characterize normal and abnormal tissues in DSC-PWI images, this study used the established optimal feature selection method *best_method* to select mRS radiomics features (mRSRF) matching 90-day mRS. Furthermore, based on CTI and mRSRF, survival features (SurvF) expressing patient survivability were constructed by the Deepsurv model [47]. In detail, CTI and mRSRF can be nonlinearly fused to obtain SurvF, which is the factor describing the 90-day mRS of patients (seen in Figure 3). This study used the Deepsurv model to implement feature fusion. The feature “Age” in CTI can be regarded as event time, while the ground truth “90-day mRS” represents the observed event. The epoch was set as 4000, the batch size was 20, and the concordance index (C-index) was used to evaluate the performance in training. The outputs were normalized according to Equation (1) to reduce the difference in features. Then, SurvF, CTI, and mRSRF were combined to obtain four types of fusion features, including CTI and mRSRF (CTI + mRSRF), CTI and SurvF (CTI + SurvF), mRSRF and SurvF (mRSRF + SurvF), and the fusion of ALL three (ALL). In this study, three groups of single features and four fusion features constitute seven.

Moreover, as seen in Table 4, to thoroughly verify the performance in various predicting conditions, we designed 90-day mRS into three situations, namely, 7-category (mRS_7), 4-category (mRS_4), and 2-category (mRS_2). The 90-day mRS 0–6 in the 7-category were scored by two experienced clinicians. The other two situations were assigned based on the number of samples and the functional outcomes indicated by mRS. In general, a higher 90-day mRS means a poorer outcome. For example, 90-day mRS 0 represents no symptoms, 6 illustrates death, and 1–5 indicates mild to severe symptoms in the 7-categories [48]. Similarly, 90-day mRS 0, 1–2, 3–4, and 5–6 indicate no symptoms and mild, moderate, and severe outcomes in the 4-category. Moreover, 90-day mRS 0–2 and 3–6 represent good and poor outcomes in the 2-category, respectively.

#### 2.2.5. Predicting Ischemic Stroke Outcome

This study used the ten machine learning models introduced in Section 2.2.3 to evaluate the performance of features. The seven groups of features, including three single types of features and four fusion features, were divided into training datasets and testing datasets with a 7:3 ratio to predict ischemic stroke outcomes. Moreover, three experiments were conducted depending on the 90-day mRS, and the Auc, Pre, Acc, F1, and Recall were the five evaluation indexes.

## 3. Results

The results are provided in four sections, including extracted significant radiomics features, selected outstanding features and best method, multidimensional fusion features, and predicting stroke outcomes. The details are shown in the following.

### 3.1. Extracted Significant Radiomics Features

In the 65,800 features computed by radiomics technology, the features with *p* < 0.05 were extracted with the *t*-test operation. As a result, 19,857 (30.2%) features with *p* = 0.009 ± 0.0135 were extracted from the original 65,800 features. In the 19,857 features, the radiomics features in the Shape group disappeared since all the features in the Shape group were insignificant (*p* > 0.05). After filtering the results of the other groups of radiomics features, Wavelet and Log-Sigma contained the most significant features of 11,612 and 5551, respectively, with *p* = 0.0091 ± 0.013 (mean ± std) and 0.01 ± 0.0138. The NGTDM group had minor significant features of 139, with *p* = 0.009 ± 0.0107. As for the other groups, there were 619, 555, 526, 436, and 419 features extracted from the GLCM, First-order, GLSZM, GLRLM, and GLDM groups, with *p* = 0.006 ± 0.0014, 0.005 ± 0.0105, 0.0066 ± 0.0104, 0.0068 ± 0.0118, and 0.006 ± 0.0114 (seen in Figure 4 and Table 5).

### 3.2. Selected Outstanding Features from Multiple Feature Selection Methods

With the 13 feature selection methods, 128 outstanding features were selected and renamed by combining the letter “F” and serial number from 1 to 128 (see Appendix A). Of all the 128 features, there were 70 Wavelet features, 2 GLDM features, 16 GLCM features, 12 First-order features, and 28 Log-Sigma features.

In the methods based on TI (seen in Table 6), 64 excellent features with *p* = 0.009 ± 0.014 were selected, where CMIM, MIM, and JMI selected 20 features, and MRMR and MIFS selected 18 features, respectively. Moreover, these five feature sets had high repeatability and consistency. For SIF, 18 features were selected, where only four came from *F_Fisher_*, and *F_LS_* and *F_ReliefF_* contributed 6 and 16 features, respectively. In contrast with *F_TI_*, the features in *F_SIF_* had lower *p*-values. In the methods based on STF, 11 features were selected, which is the lowest. These 11 features were all in *F_TS_*, and only six were included in *F_FS_*, with *p* < 0.0001. In the methods based on SSL, there were 47 selected features. The three feature sets (*F_MCFS_*, *F_Lasso_*, and *F_Alpha_*) were scattered and independent. *F_MCFS_* screened 20 features independent of *F_Lasso_* and *F_Alpha_*, while *F_Lasso_* and *F_Alpha_* shared a few.

### 3.3. Feature Sets Selected from Ten Models

Based on the tenfold cross-validation results of the ten models, the CS of the 13 *F_method_* could be computed. For the five indexes, the mean Acc (mAcc), mean Pre (mPre), mean Auc (mAuc), mean F1 (mF1), and mean Recall (mRecall) of all feature sets are 0.849, 0.851, 0.893, 0.853, and 0.872, respectively. The mAcc of each feature set on the ten models ranges from 0.71 to 0.933, while mPre, mAuc, mF1, and mRecall are from 0.706 to 0.946, from 0.768 to 0.982, from 0.724 to 0.959, and from 0.776 to 0.961. According to the average performance of feature sets in the ten models, the minimum CS is 0.738, and the maximum is 0.964 (seen in Figure 5). Specifically, taking CS as a reference, the best is *F_Lasso_* (CS = 0.964), and *F_Alpha_* achieves a comparable CS of 0.942. In contrast, *F_MRMR_*, *F_JMI_*, and *F_MIFS_* in *F_FI_* perform relatively poorly, with CS at nearly 0.740. The other feature sets scores at different levels, ranging from 0.798 to 0.938. As a result, the feature selection method Lasso is the *best_method* due to having the best CS.

### 3.4. Multidimensional Fusion Features

This section will introduce the obtained mRS radiomics features under the three situations, the performance of Deepsurv models, and the stroke outcome prediction based on the obtained seven groups of features.

#### 3.4.1. Selected 90-Day mRS Radiomics Features from *F_method_* and Generated Survival Features

For the 90-day mRS (7-category, 4-category, and 2-category), 14, 14, and 15 features with nonzero coefficients were extracted from 128 features in *F_method_* by the *Best_method* Lasso (seen in Figure 6a). Then, the combination of CTI and mRSRF was treated as input to train the Deepsurv model, and SurvF could be obtained after training. The C-index of the training model in the three situations is about 0.95 (seen in Figure 6b). The obtained SurvF is relatively concentrated, and the Pearson correlation coefficient (r-value) and *p*-value were computed and are shown in Figure 6c–e. According to the results, *p* > 0.05 for SurvF in mRS_4 and mRS_2 and mRS_7. SurvF between the other situations is statistically significant. Finally, seven feature groups were constructed by CTI, mRSRF, and SurvF for each case.

#### 3.4.2. Performance of Predicting Ischemic Stroke Outcomes

This study evaluated the predicting performance of CTI, mRSRF, SurvF, and their combination in the three classification situations. For mRS_2 (seen in Figure 7), the best Auc, Pre, Acc, F1, and Recall are 0.949, 0.969, 0.964, 0.962, and 0.964, respectively. Depending on the average of the five indexes on all the ten models, the performance of the seven feature groups from high to low is ALL, CTI + SurvF, CTI, CTI + mRSRF, mRSRF + SurvF, mRSRF, and SurvF. The combination of ALL achieves the highest mAUC of 0.884, mAcc of 0.864, mPre of 0.877, mF1 of 0.86, and mRecall of 0.864. Moreover, for a single-index Auc, the best score of mRSRF is 0.862, while for mRSRF + SurvF, it is 0.944. Similarly, CTI achieves a score of 0.928, while CTI + SurvF achieves a better score of 0.949. For the other four indexes, the seven groups show similar performance patterns such that adding SurvF will improve the outcome prediction. Furthermore, the performance of CTI + mRSRF is not stable. In detail, they are superior to CTI in models SVM, RF, NN, LR, NB, and GBDT, but the opposite results were obtained in other models.

For mRS_4 (seen in Figure 8), the best Auc, Pre, Acc, F1, and Recall are 0.908, 0.858, 0.821, 0.802 and 0.815, respectively. ALL achieves the best mAuc of 0.787, while CTI + SurvF performs best with mAcc = 0.611, mPre = 0.622, mF1 = 0.595, and mRecall = 0.611. The results that SurvF will improve the outcome prediction and clinical performance better than mRSRF are similar to their performance in mRs_2. Moreover, the combination of CTI + mRSRF performs better than CTI in models LR, NB, and DA.

For mRS_7 (seen in Figure 9), the best Auc, Pre, Acc, F1, and Recall are 0.902, 0.821, 0.75, 0.739, and 0.739, respectively. From the mean of the five indexes, CTI + SurvF performs best with mAuc of 0.788, mPre of 0.519, mAcc of 0.529, mF1 of 0.495, and mRecall of 0.47, followed by CTI, ALL, CTI + mRSRF, SurvF, mRSRF + SurvF, and mRSRF. For a single-index Auc, the best score of mRSRF is 0.644, while mRSRF + SurvF is 0.668. Similarly, CTI achieves a score of 0.891, while CTI + SurvF achieves a better score of 0.902. CTI + mRSRF achieves an Auc score of 0.843, and All achieves 0.889. The seven groups show similar performance patterns in the other four indexes such that the additional SurvF will improve the outcome prediction. In this experiment group, CTI performs better than CTI + mRSRF in all the ten models.

Moreover, for the ten models, RF, LR, NN, Ada, and GBDT achieve better performance than the others in all three classification situations. Overall, RF performs relatively stable, achieving good scores in all three situations. Although the survival features SurvF can improve the prediction of stroke outcomes in the three situations (mRS_2, mRS_4, and mRS_7) to varying degrees, due to the limitation of the small sample size, it is necessary to verify the methods further.

## 4. Discussion

Previous studies mainly used limited CTI or medical images to predict stroke outcome. Since radiomics technology can extract high-dimensional features from medical images, this study used multiple strategies to extract the outstanding temporal radiomics features of stroke lesions in DSC-PWI, used to extract the mRSRF influencing the 90-day mRS. Then, the ability of seven feature groups constructed by CTI, mRSRF, and nonlinear SurvF in the prognosis of ischemic stroke was examined by the ten learning models. Our analyses reveal that the performance of CTI is superior to mRSRF and SurvF. Furthermore, a direct combination of CTI and mRSRF can improve predictive performance on appropriate models, so model selection is needed. However, the nonlinear fusion feature SurvF improves when combined with other feature groups. For example, for mRS_2, mAuc increases from 0.874 to 0.88 when SurvF is combined with CTI. It increases from 0.872 to 0.884 when fused to CTI + mRSRF, and from 0.798 to 0.872 when fused to mRSRF. For mRS_4 and mRS_7, the same pattern can be obtained. With the survival features SurvF, this study could provide a potential clinical tool for detailed clinical predictions in ischemic stroke patients before treatment.

Although the parameter *Tmax* obtained from DSC-PWI images has been commonly used to distinguish between LT and NT, few studies have explored the value of temporal features among DSC-PWI images in predicting stroke outcomes, leaving a gap in knowledge. It has been shown that the time–intensity curve of LT in the DSC-PWI images has a much smaller brightness decrease than the curve of NT [36,37,38]. Therefore, these temporal features are correlated with the blood flow state of brain tissues to a certain extent. This study successfully extracted outstanding radiomics features to represent LT and NT with multilevel feature selection methods. The time values of the selected outstanding features were mainly concentrated at the initial moment (0–3) and the time the contrast agent passed through (17–22). In contrast, a few features were located at the end of the reaction (time greater than 30) (seen in the Appendix A). The results show that the features extracted in this study can fully express the initial state of the brain tissue, the degree of the change, and the time of the change, which are the main features for identifying abnormal blood flow in stroke patients. As a result, 128 features were selected, and their CS ranged from 0.738 to 0.964. The *best_method* Lasso achieved mAcc = 0.958, mPre = 0.96, mAuc = 0.982, mF1 = 0.959, and mRecall = 0.961 on the ten learning models. The current results prove that with the radiomics technique, the essential temporal features of the impaired tissues could be captured. It is challenging to estimate clinical outcomes only by considering the radiomics features of lesions [29]. Multiple factors are associated with the functional outcome, as well as the features of the lesion itself. Previous reports have systematically assessed ischemic stroke over 3 months and concluded that age, prior stroke, initial neurological deficit, NHISS, and lesion location are highly correlated with functional outcome [12,13,14,49]. However, few studies have performed the prediction by combining all these features. This study developed the research to compare the ability of CTI, mRSRF, and CTI+mRSRF to predict 90-day mRS. The current results show that the performance of combination features CTI + mRSRF is different among the ten models. For mRS_2 and mRS_4, it is better than single-feature CTI and mRSRF in some models, such as LR, NB, etc., but it is slightly worse than CTI in other models. Therefore, model selection is necessary when CTI + mRSRF is used to perform predictions in mRS_2 and mRS_4. Moreover, in the case of mRS_7, single-feature CTI performs better than CTI + mRSRF in almost all the ten models. The difference in the results of mRS_7 may be due to the small sample size, which requires follow-up verification.

Previous studies [50,51,52] have shown a close association between radiomics and survival in patients with tumors, chronic obstructive pulmonary disease, and cancers. The patient outcome in this study is a manifestation of survival. Therefore, it is possible to combine survival features with clinical information to verify the effect of survival features on the prognosis prediction of stroke patients. Several studies have shown that the complex and nonlinear features outputted from the Deepsurv model are superior in predicting personalized treatment recommendations compared to the state-of-the-art survival method [47,53]. Thus, this study used the Deepsurv model to generate survival features (SurvF) and verified its prediction ability by combining it with other features. The results show that SurvF improves the predictive value compared with the situation without it. By combining SurvF with CTI, the maximum increase of Auc is about 0.05 in the three situations, mRS_2, mRS_4, and mRS_7. Moreover, when fusing SurvF with mRSRF, the maximum increase of Auc is 0.08; when fusing SurvF with CTI + mRSRF, the maximum increase of Auc is about 0.09. The above study and the current results prove that early clinical information, the selected radiomics features, and the survival features can anticipate the clinical outcome. Although only a slight improvement, this study provides new implications for the risk assessment of stroke prognosis and the prediction of functional prognosis. In-depth research and improvement can be carried out on this basis. In addition, the improvement in the mRS_2 is better than in mRS_4 and mRS_7, which may cause by the small sample size. Due to the small samples, the number of samples under each category will decrease when categories increase, leading to insufficient training. If there is enough sample size in the future, we will verify them further.

Furthermore, we would like to discuss the outcome prediction ability of CTI, mRSRF, and SurvF. For CTI and mRSRF, the information in CTI includes basic patient information, symptoms, treatment strategies, clinical score, and lesion size. mRSRF includes the dynamic blood flow state information extracted from the DSC-PWI image. When comparing CTI and mRSRF separately, CTI performs better than mRSRF due to the comprehensive overall features of the patient, while mRSRF is only a single image feature. This result has been proved in previous research and this study. However, when combining CTI and mRSRF, a linear or nonlinear relationship can be established between the clinical characteristics, blood flow state features, and functional recovery of patients, which will improve the outcome prediction. However, as shown in Figure 7, Figure 8 and Figure 9, the prediction ability of the combination of CTI and mRSRF is relatively unstable. Therefore, a new survival feature SurvF generated from a widely used survival model Deepsurv is proposed. To some extent, SurvF can represent the survival probability of patients under the condition of age based on the CTI and mRSRF. Thus, the outcome prediction ability can be improved when the feature SurvF, which can represent survivability, is added to CTI, mRSRF, and their combination.

There are some limitations of this study. First, the size of the datasets is relatively small, and all data came from a single hospital, which may have led to biased results and a lack of generalizability. To address the limitation, we segmented LT from the DSC-PWI images and defined the normal tissue in the symmetrical areas of LT as NT. This way, one group of LT and NT could be generated from one patient. The double samples (160) could be obtained from 80 DSC-PWI images, and the positive and negative sample sizes were equal. The expanded balanced samples will help select the accurate radiomics features and reduce the sample imbalance. When evaluating the performance of the selected feature sets, the tenfold cross-validation was performed to reduce the influence of sample size. In addition, ten machine learning models were used to compute the composite scores (CS) to obtain reliable results. Then, the most outstanding lesion radiomics features and the best feature selection method could be selected. For CTI, mRSRF, SurvF, and their combination in the ability of outcome prediction, limited by the small sample size, this study compared the average results of five indicators, Auc, Pre, Acc, F1, and Recall, to obtain a relatively fair result. Moreover, although the results showed that CTI + mRSRF and the fusion of SurvF with other features can improve the prediction value, further optimization of the models can be regarded as one future work. Furthermore, although several methods in our study were used to address the limitation listed, further verification is still needed. We will validate our improved method with the larger and more varied datasets before applying it to clinical trials in future work.

Finally, the results in this study do not mean that the models can be used alone for stroke treatment decision making. Instead, this study should be considered a support tool in stroke treatment guidance.

## 5. Conclusions

In conclusion, this study provides new insights into prognosis prediction in ischemic stroke. First, the results indicate that the selected radiomics features from DSC-PWI can accurately distinguish between normal and stroke tissues. Second, compared with radiomics features of stroke lesions, the clinical text information could better predict the neurological recovery of stroke patients. However, combining radiomics features and clinical text information can improve predictive performance on appropriate models, so model selection is needed. Moreover, the additional survival features generated from clinical text features and radiomics features will improve the prediction effect compared with the original features. This study could provide a potential clinical tool for detailed clinical predictions in ischemic stroke patients before treatment. However, limited by the small sample size, further validation on the larger and more varied datasets is necessary.

## Figures and Tables

**Figure 1 diagnostics-12-01664-f001:**
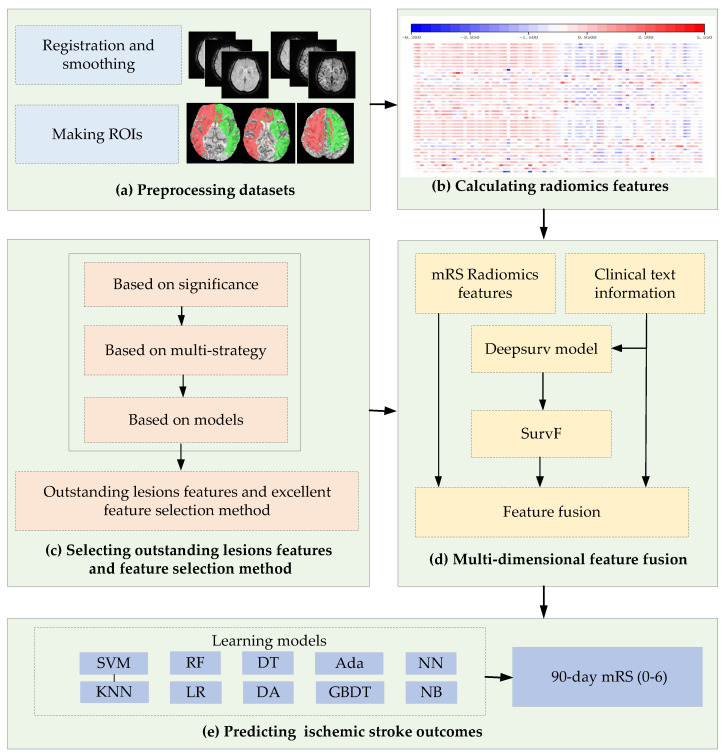
Flowchart of this study. (**a**) Process of the preprocessing of DSC-PWI datasets and making ROIs; (**b**) Calculating radiomics features, where the value of the feature is represented by color; (**c**–**e**) The process of feature selection, feature fusion, and stroke outcome prediction.

**Figure 2 diagnostics-12-01664-f002:**
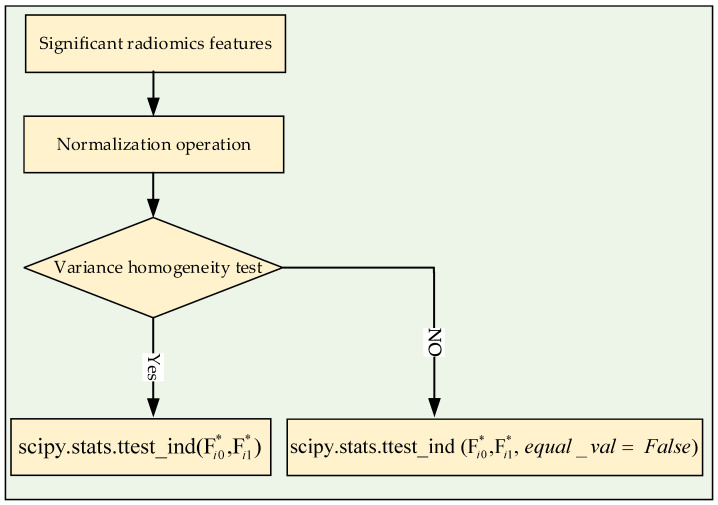
Flowchart of selecting significant features. $F_{i0}^{*}$ and $F_{i1}^{*}$ are the i*th* feature in LT and NT groups, respectively.

**Figure 3 diagnostics-12-01664-f003:**
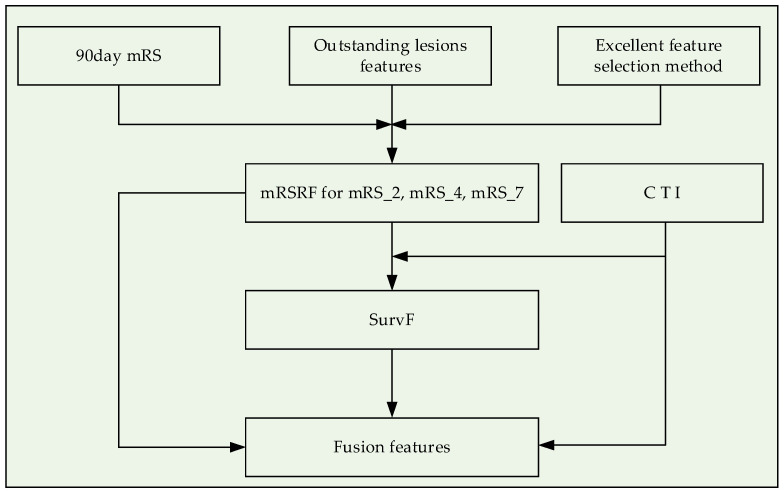
Flowchart of multidimensional feature fusion.

**Figure 4 diagnostics-12-01664-f004:**
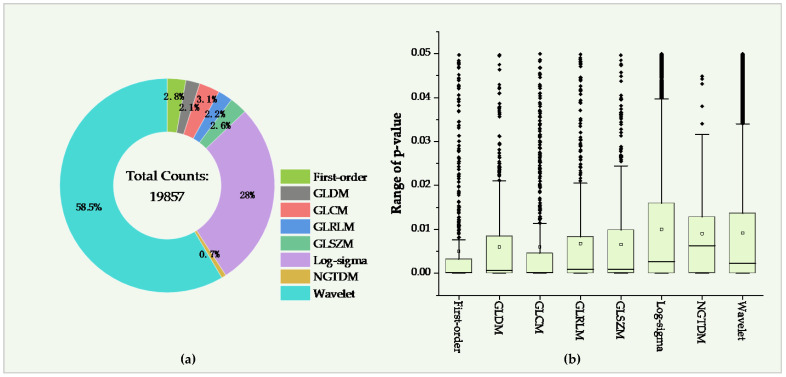
Significant features in the radiomics features group. (**a**,**b**) Ratio and *p*-value of significant features in the eight radiomics feature groups, respectively.

**Figure 5 diagnostics-12-01664-f005:**
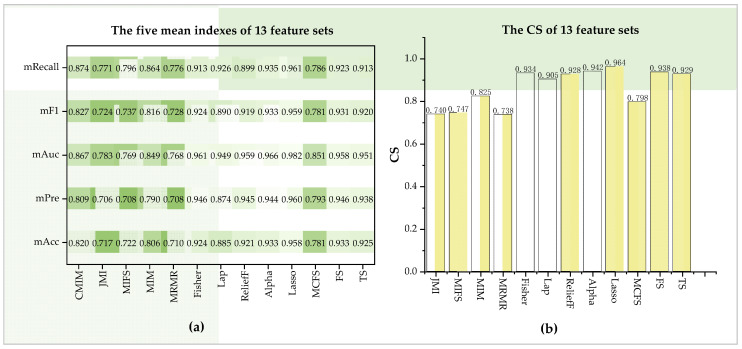
Performance of the 13 feature sets on the ten learning models. (**a**) Five mean indexes of 13 methods on the ten models and (**b**) CS of 13 methods.

**Figure 6 diagnostics-12-01664-f006:**
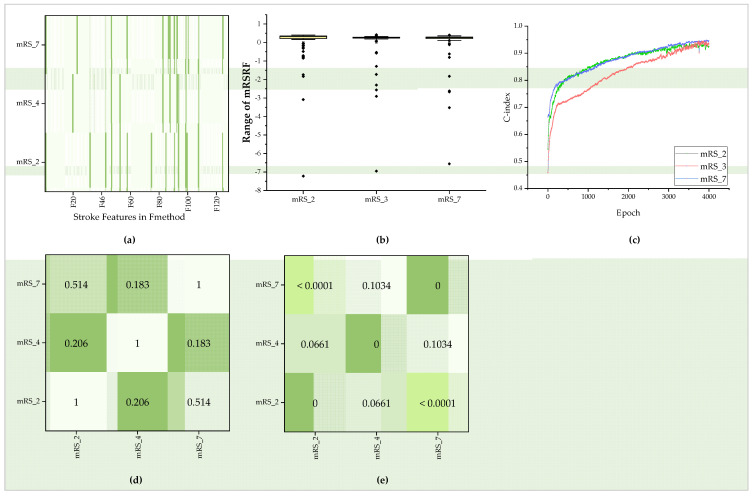
Selected mRSRF and statistics in mRS_2, mRS_4, and mRS_7. (**a**) mRSRF in three situations, and green color represents selected items from 128 outstanding features in F_method_. (**b**) Box plot among the three groups of mRSRF. (**c**) C-index of the Deepsurv model in training. (**d**,**e**) Pearson correlation coefficients and *p*-values among the three groups of mRSRF.

**Figure 7 diagnostics-12-01664-f007:**
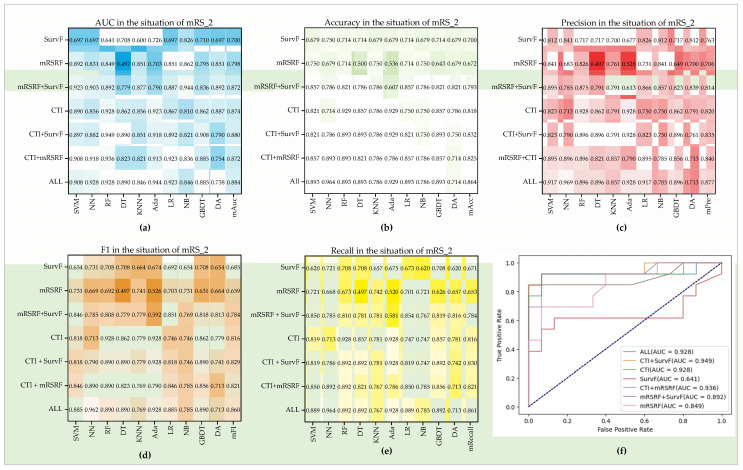
Performance of seven feature groups in the situation of mRS_2. (**a**–**e**) Auc, Pre, Acc, F1, and Recall on the ten models. (**f**) ROC curves of seven feature groups on the RF model.

**Figure 8 diagnostics-12-01664-f008:**
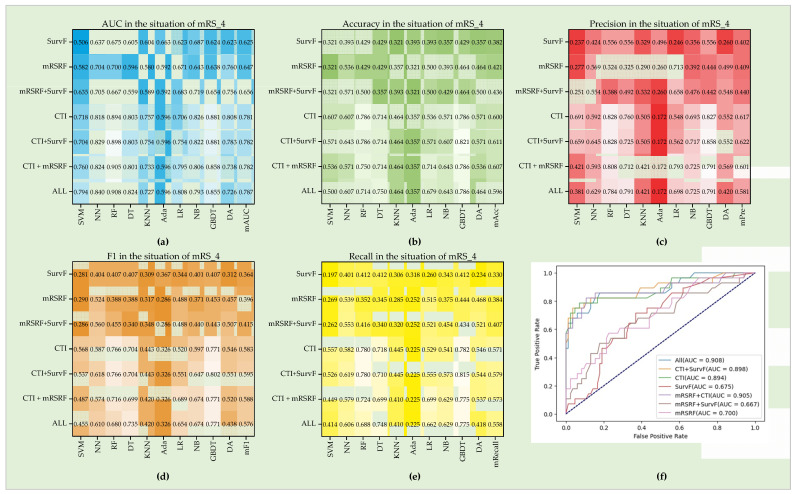
Performance of seven feature groups in the situation of mRS_4. (**a**–**e**) Auc, Pre, Acc, F1, and Recall on the ten models. (**f**) ROC curves of seven feature groups on the RF model.

**Figure 9 diagnostics-12-01664-f009:**
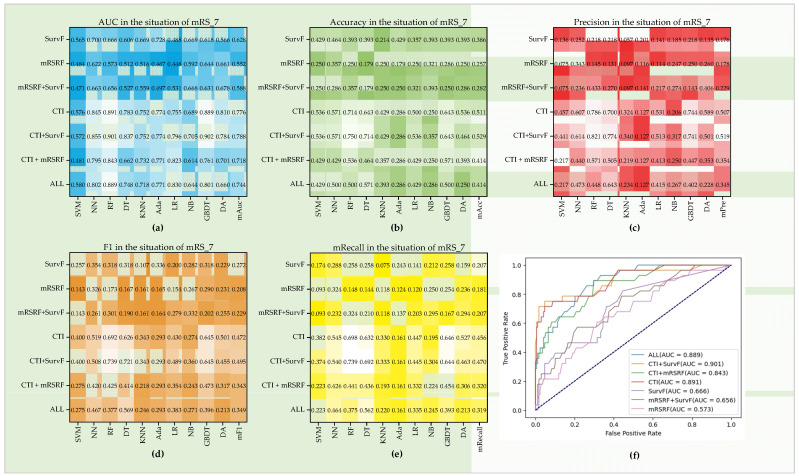
Performance of seven feature groups in the situation of mRS_7. (**a**–**e**) Auc, Pre, Acc, F1, and Recall on the ten models. (**f**) ROC curves of seven feature groups on the RF model.

**Table 1 diagnostics-12-01664-t001:** Patient information and scanning parameters of DSC-PWI datasets.

Patient Information		Scanning Parameters of DSC-PWI Images	
Numbers of patients	56	TE/TR	32/1590 ms
Datasets (sets)	80	Matrix	256 × 256
Numbers of Female (%)	15 (26.79%)	FOV	230 × 230 mm^2^
Age (Mean ± Std)	71.362 ± 10.91	Thickness	5 mm
Volumes of lesions (Mean ± Std, mL)	95.583 ± 72.304	Number of measurements	50
Income NHISS (Mean ± Std)	9.919 ± 6.747	Spacing between slices	6.5 mm
Outcome NHISS (Mean ± Std)	6.275 ± 6.875	Pixel bandwidth	1347 Hz/pixel
Right limbs weakness (%)	38 (47.5%)	Number of slices	20
Left limbs weakness (%)	36 (45%)		
Lisp out (%)	59 (73.75%)		
Confused (%)	10 (12.5%)		
Hypertension (%)	59 (73.75%)		
Diabetes (%)	26 (32.5%)		
Atrial fibrillation (%)	28 (35%)		
Intra-arterial thrombectomy (%)	22 (27.5%)		
90-day mRS	2.525 ± 2.326		

**Table 3 diagnostics-12-01664-t003:** Descriptions of the 10 models in this study.

No.	Model	Definition in Python 3.6
1	SVM	sklearn.svm.SVC(kernel = ‘rbf’,probability = True)
2	DT	sklearn.tree. DecisionTreeClassifier()
3	Ada	sklearn.ensemble.AdaBoostClassifier()
4	NN	sklearn.neural_network. MLPClassifier (hidden_layer_sizes = (400, 100), alpha = 0.01, max_iter = 10000)
5	RF	sklearn.ensemble.RandomForestClassifier(n_estimators = 200)
6	KNN	sklearn.neighbors. sklearn.neighbors()
7	LR	sklearn.linear_model.logisticRegressionCV(max_iter = 100,000, solver = “liblinear”)
8	DA	sklearn.discriminant_analysis.()
9	GBDT	sklearn.ensemble.GradientBoostingClassifier()
10	NB	sklearn.naive_bayes. GaussianNB()

**Table 4 diagnostics-12-01664-t004:** Distributions of 90-day mRS in the three situations.

90-Day mRS	0	1	2	3	4	5	6
7-category counts	25	11	9	4	8	9	14
4-category counts	25	20	12	23	-	-	-
2-category counts	45	35	-		-	-	-

**Table 5 diagnostics-12-01664-t005:** Statistics of *p*-values of significant features in each radiomics group.

Groups	Numbers	Mean	Std	Min	Medium	Max
First-order	555	0.005	0.0105	<0.0001	<0.0001	0.0497
GLDM	419	0.006	0.0104	<0.0001	<0.0001	0.0497
GLCM	619	0.006	0.0114	<0.0001	<0.0001	0.0499
GLRLM	436	0.0068	0.0118	<0.0001	<0.0001	0.0498
GLSZM	526	0.0066	0.0104	<0.0001	<0.0001	0.0496
Log-Sigma	5551	0.01	0.0138	<0.0001	0.0027	0.05
NGTDM	139	0.009	0.0107	<0.0001	0.0063	0.0449
Wavelet	11612	0.0091	0.013	<0.0001	0.0022	0.05
Shape						

**Table 6 diagnostics-12-01664-t006:** Statistics of extracted features from the 13 methods.

Type	Method	Counts of Features	*p*-Value
TI	CMIM	20	0.004 ± 0.011
	MIM	20	0.008 ± 0.015
	JMI	20	0.014 ± 0.016
	MRMR	18	0.009 ± 0.01
	MIFS	18	0.006 ± 0.009
SIF	Fisher	4	<0.0001
	ReliefF	12	<0.0001
	LS	6	0.013 ± 0.018
STF	FS	7	<0.0001
	TS	11	<0.0001
SSL	Alpha	11	0.006 ± 0.014
	Lasso	16	<0.0001
	MCFS	20	0.011 ± 0.012

## Data Availability

The data supporting this study’s findings are available from the corresponding author upon reasonable request.

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
