# Peer review of "Novel Survival Features Generated by Clinical Text Information and Radiomics Features May Improve the Prediction of Ischemic Stroke Outcome"

_diagnostics, 2022, doi:10.3390/diagnostics12071664_

Round 1

Reviewer 1 Report

This paper studies Novel Survival Features Combined with Clinical Text Information and Radiomics Features for Predicting Ischemic Stroke  Outcome based on Machine Learning Models. It is so interesting topic and related to the journal topic. There are a few weaknesses that should be addressed in this paper. Therefore, I suggest the authors resubmit it after a minor revision. My suggestions are as follows:

  1. As the first step, I strongly suggest that the paper be proofread and reread meticulously again, particularly in regard to the spelling and grammatical mistakes.
  2. The paper should be revised to include recent references on stroke and machine learning in particular 2021-2022.
  3. Some of your figures are pictures, and they are not visible. They should be in an editable and more visible version.
  4. In some cases, such as figures 2 and 3, the reference to figures is forgotten.
  5. I suggest that you update section 3.1  so that the illustration used in this section should be more readable. In fact, extracted significant radionics features part needs more explanation and clarification.
  6. It is necessary to include additional information in section 2.2.3 part in the selecting outstanding lesions features and excellent feature selection method part.
  7. In section 2, you provided a flowchart to explain your approach. This section must provide a concise and clear explanation of the suggested approach and ten machine learning approaches. Although the flowchart is beneficial, it’s also important to outline the methodology behind this new approach.
  8. In figures 5-7 you have provided 6 figures which are not visible. Please divide them and explain them separately step by step or increase the quality of the pictures.
  9.  Your conclusion is too short. Please convert some parts of the discussion into the conclusion. 

Reviewer 2 Report

The paper proposes to explore if information hidden in dynamic perfusion series images parameters can contribute to predict functional outcome in ischemic patients, when combined with clinical text information using machine learning.

The idea is legitimate and the authors do a thorough job of detailing the methological approach used to select radiomics features and combine various machine learning models for prediction. The results are not only presented for 2 categories but also for multiple categories, hoping to provide better insight into disease outcome.

The main concern with the study is complexity of the prediction process that leaves the reader facing a blackbox machinery with little information on why the resulting model should be trusted and what features contribute to the improved predictions. A paper organisation with most of the implementation details moved to supplementary and a more extensive discussion on what makes the model perform well would likely improve the readability.

It is also not clear in what regard the model indeed improves the state-of-the-art as the metrics suggest only a limited improvement for the 2-categories prediction. For the multi-class prediction, the performance is strongly degraded, questionning the usefullness of these versions.

While the idea to extract features from the longitudinal perfusion data is innovative, does it necessarilly need to replace the static perfusion parameters used in other models, or could it complement it instead? Have the authors investigated that point?

As stated in the limitations, the study suffers from a lack of external validation. Single institutions studies with limited number of cases always bear the risk of strongly overfitting to the data, hence the generalization to other institutions is likely to suffer from significant degradation in performance.

In the end, the study fails to explain why the proposed radiomics analysis can benefit the prediction results in a model hardly explainable and for which generalization is questionnable, thus asking the question of its clinical value. If the study was intended as a proof-of-concept, it should at least explain how and why it is improving the state-of-the-art.

As a side note, the paper makes use of a large number of abbreviations for which full text development is often missing, making the paper hard to read for readers not so familiar with the field.

Reviewer 3 Report

This manuscript presents a thorough study on stroke outcome prediction using deep learning models. The authors conducted a sophisticated feature extraction and selection process to pick the best feature group and then tested on 10 popular models. The paper claimed a high accuracy of prediction using the methods described. This work is of great importance to the field of medical image computing and diagnosis. The paper is well written and covers all the technical details. I have no further issue with the methods and results, just minor comments on English writing and figure presentation:

  • The Introduction: (lines 48 & 53 & 63) The authors may want to replace "manners" with "approaches"/"methods"/"treatments."
  • The Introduction: (line 50) "but undoubtedly" sounds odd. The authors could just use "but" or "nevertheless."
  • The Introduction: (line 53) "foreseen" ---> forecast
  • The Introduction: the authors need to work on the references in paragraphs 2 & 3.
  • The Introduction: (lines 69-86) the authors should include a list or a note to specify all the abbreviations used in this paragraph.
  • The Introduction: (line 98) "radiomics" ---> radiomic
  • Figure 1: the authors should explain the relationships between panel (c) and other panels. No arrows come out or go into this panel, thus making the 'flowchart" not a flow-chart.

Round 2

Reviewer 2 Report

Thank you for editing the manuscript and adressing the concerns expressed. The revised manuscript does show improvements in comparison to the original version, notably with an improved discussion detailing the relative benefits of CTI, mRSRF and survival features predicted by SurvF.

Unfortunately, I believe the study is still suffering from a too small dataset to address the challenging questions asked, notably the possibility to reliably provide multi-class predictions. The multiplication of features selection methods and predictive models and the averaging of quality metrics does little to address the shortage of data and the plausible overfitting problem that can result from it.

I am also still questionning the clinical acceptance of such a predictive model where complex and sometimes obscure radiomics features are used to predict clinical outcome, with little explaination in the paper given to what the features retained mean and how they relate radiologically or physiologically to ischemic stroke.
